# Epstein–Barr Virus and Clinico-Endoscopic Characteristics of Gastric Remnant Cancers Compared to Proximal Non-Remnant Cancers: A Population-Based Study

**DOI:** 10.3390/cancers16112000

**Published:** 2024-05-24

**Authors:** Erling A. Bringeland, Christina Våge, Ann A. S. Ubøe, Alina D. Sandø, Patricia Mjønes, Reidar Fossmark

**Affiliations:** 1Department of Clinical and Molecular Medicine, Norwegian University of Science and Technology (NTNU), 7030 Trondheim, Norway; chrivaa@stud.ntnu.no (C.V.); aauboe@stud.ntnu.no (A.A.S.U.); alina.desiree.sando@stolav.no (A.D.S.); patricia.mjones@ntnu.no (P.M.); reidar.fossmark@ntnu.no (R.F.); 2Department of Gastrointestinal Surgery, St. Olav’s Hospital, 7030 Trondheim, Norway; 3Department of Pathology, St. Olav’s Hospital, 7030 Trondheim, Norway; 4Department of Gastroenterology, St. Olav’s Hospital, 7030 Trondheim, Norway

**Keywords:** gastric remnant cancer, Epstein–Barr virus, survival

## Abstract

**Simple Summary:**

Epstein–Barr virus (EBV)-associated gastric cancer is accepted as a distinct entity among gastric adenocarcinomas. It is particularly frequent in gastric remnant cancer (GCR), defined as cancer in the gastric stump several decades after distal gastric resection for benign disease. Gastric cancer in the West is a comparatively rare disease, and most of the published data stem from the East. Using population-based data from the West spanning the years from 2001 to 2016, we were able to show that a significantly higher proportion of GRCs compared to non-GRC proximally located cancers were EBV-positive, even in the West. The mode of presentation and findings at upper endoscopy were more subtle and differed significantly from proximally located non-GRC cancers, possibly making the GRCs more difficult to diagnose.

**Abstract:**

Epstein–Barr virus (EBV) is associated with 5–10% of gastric cancers and is recognized as a distinct molecular subtype. EBV positivity is particularly high in gastric remnant cancer (GRC), which may inform the mode of clinical presentation and findings at endoscopy. Most data are from the East, and the question remains how this applies to a Western cohort. We conducted a population-based study in Central Norway, 2001–2016. Patients with GRC (n = 78) and patients with non-GRC proximally located cancer and available tissue for EBV status (n = 116, control group) were identified from the Norwegian Cancer Registry. Relevant data were collected from the individual patient journals. EBV status was assessed using in situ hybridization. The median latency time from the distal gastrectomy to GRC was 37.6 (range 15.7–68.0) years. GRC more often presented with GI bleeding, 31.0% vs. 16.1%, *p* = 0.017, and at endoscopy more seldom with an ulcer, 19.7% vs. 38.2%, *p* = 0.012, or a tumour, 40.8% vs. 66.4%, *p* < 0.001. For GRC, 18.7% were EBV-positive compared to 6.0% among the controls, *p* = 0.006. EBV status was not associated with patient age, sex, or Lauren histological type. No difference in long-term survival rates between GRC and controls was found or between EBV-positive vs. -negative GRCs. In conclusion, a higher proportion of GRC cases, compared to controls, are EBV positive, indicating different causative factors. The mode of clinical presentation and findings at endoscopy were more subtle in the patients with GRC.

## 1. Introduction

The first report of Epstein–Barr virus (EBV)-associated gastric malignancy (lympho-epithelial carcinoma) was published in 1990 by Burk et al. [1]. Two years later, this was corroborated for gastric adenocarcinomas [2]. It was demonstrated how the EBV genome resides in gastric cancer cells and the surrounding dysplastic epithelium, leaving the normal gastric mucosa unaffected, thus adding to the list of EBV-associated malignancies [3]. In 2014, EBV-positive gastric cancer was acknowledged as a distinct molecular subtype in the Cancer Genome Atlas Network [4]. The reported frequency of EBV positivity in gastric adenocarcinomas features great variation, and, in a recent meta-analysis, a pooled prevalence of 8.8% (95% CI 7.7–9.9) was estimated in a random-effects model [5]. The mechanism through which the EBV inflicts gastric carcinoma is still a matter of debate. The majority of the adult population has serological evidence of previous EBV infection as demonstrated by the presence of IgG antibodies, yet only a minor proportion proceeds to develop EBV-associated gastric cancer. One main theory suggests a mechanism whereby saliva, hosting infected B-lymphocytes and epithelial cells, reaches the stomach and directly contributes to infection. Another theory advocates that the reactivation of dormant infected B-cells buried in the gastric mucosa may occur, thereby releasing the oncogenic EBV genome into the gastric cells, as recently summarized in a review by Sun et al. [6]. The result is a carcinoma that is formed by the monoclonal proliferation of EBV-positive tumour cells.

The propensity for adenocarcinoma to develop in the remainder of the stomach following partial gastrectomy was described more than a century ago [7] and coined with the name gastric remnant cancer (GRC). The definition has by some authors been restricted to cancers following distal gastrectomy for benign disease, whereas malignant disease has been included by others [8]. In a meta-analysis, the risk appeared to be independent of whether a Billroth I or a Billroth II reconstruction had been performed [9]. The relevance of GRC as a separate entity has been questioned since some disease characteristics and prognoses do not seem to differ from gastric cancers in general [10,11]. The aetiology of GRC, however, may differ from that of non-GRCs.

Long-term exposure of the gastric mucosa to bile acids due to pronounced duodeno-gastric reflux is perceived to have a carcinogenic effect [12,13]. Gastro-jejunostomy was proven carcinogenic in a rat model, a risk further enhanced by proton pump inhibitor-induced hypoacidity and hypergastrinemia [14]. Conceivably, chronic bile reflux gastritis subsequent to a Billroth I or II reconstruction may increase mucosal vulnerability and facilitate EBV invasion of the gastric epithelium [15,16]. A markedly higher proportion of GRCs than non-GRCs was found to be EBV-positive in a recent meta-analysis of mainly Eastern patient cohorts [17]. In addition, symptoms and endoscopic findings at diagnosis may differ between GRC and non-GRC gastric cancers, thereby justifying GRC as a distinct clinical entity.

This study aimed to assess the prevalence of EBV infection in GRC tissue as well as clinical and endoscopic characteristics in patients with GRC in a Western population and to contrast the findings to patients with a non-GRC proximal gastric cancer in the same population. To the best of our knowledge, no such population-based study from a Western cohort has been published during the last three decades.

## 2. Materials and Methods

### 2.1. Study Design and Data Source

We have previously reported population-based data from central Norway on patients diagnosed with gastric cancer (n = 1217), including Siewert types II and III, from 2001 to 2016 [18,19]. In short, the patients were identified by a combined search in the Norwegian Cancer Registry (NCR) and the Norwegian Patient Register (NPR) databases. By using the 11-digit identification number unique to each citizen, the individual electronic patient journals (EPJs) were reviewed, and patients with diagnoses other than adenocarcinoma could be excluded. For the remaining patients, relevant information including survival data were obtained from the EPJs. The present study was concerned with the subset with GRC and the aspects relating to EBV positivity.

Gastric adenocarcinoma in patients with previous distal gastrectomy due to either benign or malignant disease was defined as GRC, n = 78 (6.4%). Demographic data, indication for the index distal gastrectomy (benign or malignant disease), the date of this resection, and the method of reconstruction, have recently been published [20]. Based on estimates of EBV positivity in Eastern patient cohorts [17] and power calculations, n = 116 patients among the proximal non-GRCs with available tissue blocks for EBV histopathological analyses were randomly selected to serve as the control group (SPSS random case selection process). Proximal non-GRCs were chosen, since differences in the rate of EBV positivity between distal and proximal gastric cancers are well known [5,15,17] and since the aetiology and risk factors of gastric cancers also differ between locations within the stomach [21,22]. The EPJs were then revisited, and the following additional information was extracted for the purpose of this study: the indication for referral to upper endoscopy for the present cancer diagnosis and the findings at endoscopy. A MAGIC-style regimen of perioperative chemotherapy was since 2007 offered to medically fit patients under 75 years with clinical stage I–III disease [23]. Resection surgery included by default a modified D2 lymphadenectomy, and, for patients with GRC, a gastrectomy with esophago-jejunostomy Roux-Y reconstruction was implicit.

Censoring day was 1 February 2023, allowing for a minimum follow-up of 6 years and 7 months.

### 2.2. Histopathology and EBV In Situ Hybridization

Histological sections stained with haematoxylin and eosin were reviewed from all patients by an experienced pathologist (PM), and the carcinomas were classified according to Lauren [24]. EBV in situ hybridization (ISH) was performed on sections cut from formalin-fixed paraffin-embedded tissue. Sections were deparaffinized and underwent enzyme treatment with ISH Protease 3 (Catalogue number 780-4149, Roche Diagnostics GmbH, Mannheim, Germany). The probe against EBV-encoded RNA (INFORM EBER, Epstein–Barr Virus Early RNA probe, kat.nr: 800-2842, Roche, Basel, Switzerland) was used, and the complex was visualized (VENTANA ISH iVIEW_Blue_ Detection Kit, kat.nr: 800-092, Roche). Sections were counterstained (Red Counterstain II, kat.nr: 780-2218, Roche) and examined by an experienced pathologist (PM).

### 2.3. Statistical Analyses

Continuous variables were summarized using the median (range) and compared using the Mann–Whitney U test. Categorical variables were tabulated and analysed by the chi-square test. Survival curves were constructed with the Kaplan–Meier method and compared using the log-rank test. A *p*-value < 0.05 was considered significant. Statistical analyses were performed using SPSS version 29.0.1 (IBM, Armonk, NY, USA).

### 2.4. Ethics Approval

The gastric cancer projects were approved by the Regional Committee for Medical and Health Research Ethics (2011/1436 and 2016/2173).

## 3. Results

### 3.1. Patients Demographics, Tumour-, and Treatment Variables

As previously reported, the median time elapsed from the index distal gastrectomy to the occurrence of GRC was 37.6 (15.7–68.0) years [20]. Of the 78 patients with GRC, 76 (97.4%) had their index operation for benign peptic ulcer disease and two (2.6%) for malignant disease. Information on the method of reconstruction was available in 73 patients, with 64 (87.7%) ad modum Billroth II, 8 (11.0%) ad modum Billroth I, and one patient in a Roux-Y configuration [20]. Patients with GRC were older than patients with proximal non-GRC, median 79 years vs. median 72 years, *p* = 0.005 (Table 1). The proportion of men was higher among patients with GRC, 83.3% versus 69.8%, *p* = 0.033. The Lauren distribution, surgical treatment offered, and (y)pTNM-stage distribution did not differ significantly between GRC and proximal non-GRC patients. A significantly lower proportion of patients with GRC, however, received perioperative chemotherapy or palliative chemotherapy at any time, likely due to the significantly higher median age (Table 1).

### 3.2. EBV Status

Sufficient tumour tissue for EBV in situ hybridization (ISH) was available from 75 of 78 GRC tumours. Of these, 32/75 (42.7%) of the EBV-ISHs were carried out on surgical specimens, the remaining on biopsies only. The corresponding proportion in the control group was 48/116 (41.4%). Typical examples of EBV positivity, as expressed in a Lauren diffuse and in a Lauren intestinal-type cancer along with a negative control, are depicted (Figure 1). In EBV-positive tumours, most of the tumour cells stained positive for EBV. The overall proportion of EBV-positive tumours was 18.7% (n = 14) for GRCs and 6.0% (n = 7) for proximal non-GRCs, *p* = 0.006 (Table 2), OR 3.6 (95% CI 1.4–9.3). No significant differences were found in the proportions of EBV-positive tumours, stratified by the Lauren distribution, age category, or sex (Table 2). None of the tumours had the histological appearance of a lympho-epithelial carcinoma.

### 3.3. Symptoms at Time of Diagnosis and Findings at Upper Endoscopy

Only 15/78 patients with GRC (19.2%) had received an upper endoscopy between the index distal gastrectomy and the current GRC diagnosis. Four patients with GRC and five proximal non-patients with GRC had their cancer diagnosed at a surveillance endoscopy (Table 3). As a symptom at diagnosis, gastrointestinal bleeding was more common among the patients with GRC than among controls, with 31.0% vs. 16.1%, *p* = 0.017, whereas abdominal pain was less common 14.1% vs. 35.7%, *p* < 0.001. At upper endoscopy, both the finding of either an ulcer, 19.7% vs. 38.2%, *p* = 0.012, or a tumour, 40.8% vs. 66.4%, *p* < 0.001, was less common among the patients with GRC than the controls (Table 3). For the remaining categories of symptoms or findings at endoscopy, no significant differences were found.

### 3.4. Long-Term Survival Rates

The median overall survival for patients with GRC was 7.5 months (95% CI 3.8–11.1) compared to 8.8 months (95% CI 7.0–10.7) for those with proximal non-GRC patients, *p* = 0.831. Survival curves are depicted in Figure 2a, log-rank *p* = 0.801. The GRC patient cohort was then split by EBV status. The fourteen EBV-positive GRCs had a median survival of 22.2 months (95% CI 0.0–64.9) compared to 7.5 months (95% CI 5.1–9.8) for the larger EBV-negative subset, *p* = 0.240. Survival curves are depicted in Figure 2b, log-rank *p* = 0.591.

## 4. Discussion

In this population-based cohort of gastric adenocarcinomas from the West diagnosed from 2001 to 2016, GRCs accounted for 6.4% of the patients, falling within the upper end of the range of reported values [25,26]. The GRCs were diagnosed after a median latency of 37.6 years following the index distal gastrectomy, which compares well to that reported by others [10,26,27,28]. Only 2/78 GRC cases had an index operation for gastric malignancy.

### 4.1. EBV-Positive Cancers

The previously reported proportions of EBV-positive gastric cancer lie within a wide range [5]. This may reflect true variations but could also be influenced by the method used to detect the EBV and whether surgical specimens vs. limited biopsies were examined [5]. Furthermore, the cost-efficient polymerase chain reaction (PCR) has a higher sensitivity but a low specificity compared to the in situ hybridization method (ISH), which is considered the gold standard and was used in the present study [5,6]. For the subset of GRC in particular, data on EBV positivity are scarce. To the best of our knowledge, no population-based study from a Western cohort with proximally located non-GRC cancers as the control group has been published during the last thirty years.

The main finding in the present study was that 18.7% of the GRCs were EBV-positive, 3.1 times higher than the proportion in the control group of proximal non-GRC, OR 3.6 (95% CI 1.4–9.3). The finding is consistent with those of a recent meta-analysis, which reported OR 5.2 (95% CI 3.9–7.0) [17]. Notably, these were almost exclusively patients from Eastern patient cohorts, which may not be representative of a Western GRC population. Furthermore, the OR was estimated based on GRC compared to gastric cancer in general. Several studies report, however, a higher proportion of EBV positivity in tumours with a proximal location. Tumours in the gastric cardia or corpus are reported to be positive at least twice as often compared to tumours arising in the gastric antrum [5,15,29]. Hence, by choosing an anatomically similar non-GRC control group, as in the present study, a somewhat higher EBV positivity and lower OR were to be expected.

A larger proportion of gastric cancer in men compared to women has been reported to be EBV-positive [5,15], even for the subset of GRC [25]. In the meta-analysis by Tavaloki et al. on gastric cancer in general, a 1.9-fold-higher prevalence of EBV positivity in male patient tumours was found (10.8% vs. 5.7%, respectively, *p* < 0.0001) [5]. In the present study, including only proximal cancers, similar numbers were reproduced, although without statistical significance, with 12.5% EBV positivity in men vs. 6.4% in women, *p* = 0.244. For histologic category, the meta-analysis by Tavaloki et al. found a prevalence of EBV positivity of 8.1% in Lauren intestinal tumours and 9.4% in Lauren diffuse tumours, *p* = 0.31. Although the numbers are small, similar proportions were found in the present study restricted to proximal cancers, i.e., GRC and non-GRC proximal cancers (Table 2).

### 4.2. Demographics

Men are more often than women diagnosed with gastric cancer, and a ratio approaching 2:1 is typical in a Western population [18]. For the GRC cohort, an even higher proportion of 83.3% were men in the present study, consistent with the value in previous reports [28,30]. Although the mechanisms that fully explain this predominance in GRC are wanted, a certain contributing factor is that men were more often subjected to distal gastric resection due to recurrent ulcer disease than women decades ago [27,30]. A median age of 79 years at the time of diagnosis for patients with GRC is significantly higher than that in the control group. This has also been acknowledged by several others [31,32] and reflects the long latency time for GRC to develop.

### 4.3. Symptoms at Diagnosis and Endoscopic Findings

Norwegian national guidelines have not recommended surveillance endoscopies following surgical treatment for gastric cancer [33]. This is at variance with Japanese tradition, offering surveillance gastroscopies for at least ten years after surgery [34]. For benign disease, in case of a lack of solid evidence, experts have pragmatically suggested surveillance endoscopy after distal gastrectomy to start 15–20 years after surgery [9,25]. Only 19.2% of the patients with GRC in the present study had received an upper endoscopy between the distal gastrectomy and the GRC diagnosis. The large majority of the patients with GRC was referred to upper endoscopy due to alarm symptoms.

Abdominal pain was significantly less common in patients with GRC than in patients with proximal non-GRC, whereas other symptoms, such as dyspepsia, vomiting, or weight loss, did not differ. The observed differences may relate to that patients with GRC, having previously undergone partial gastrectomy, misinterpret and underreport symptoms [25]. However, it must be mentioned that variables such as symptoms in particular are prone to bias in a retrospective study. At diagnostic endoscopy, GRCs less often appear as an ulcer or a frank tumour, and more subtle tumour manifestations could impose a false impression of non-malignant disease. Others have reported that EBV-associated non-GRC often have a submucosal tumour-like (non-ulcerated) appearance [35], and directed biopsies after long-standing distal gastric resections have been advocated [25].

### 4.4. Long-Term Survival Rates

The prognosis for patients with GRC does not seem to differ significantly from that of gastric cancer patients in general [10,25,27,36] or from those with proximally located cancers in particular [11], as also found in the present study. It has been pointed out, though, that small sample sizes and a variety of national and genetic backgrounds could confound conclusions [10]. Keeping the comparatively small population size in mind, there were no significant differences in survival between the patients with GRC and the control group, or across the EBV status in the GRC group in the present study. In the near future, matters may look different, with immunotherapy emerging as a new angle of attack for EBV-positive cancers [37,38].

A Cox proportional hazard model with, e.g., age, sex, tumour location, disease stage, and EBV status as explanatory variables could be envisaged. We do believe, however, that since several of the variables mentioned might be mediators of risk rather than confounders, neutralising them in a multivariable analysis would not be informative.

### 4.5. Strengths and Limitations

The strengths of the study include its population-based design. A high proportion of patients with GRC had available tumour tissue for EBV analysis, which was conducted using the ISH method, acknowledged for both high sensitivity and specificity. Furthermore, the follow-up was complete, and patient records including endoscopy reports were available for almost all patients. Limitations include the retrospective nature of this study, rendering the variables “symptoms at diagnosis” and “findings at upper gastroscopy”, in particular, to be registered with less precision. Furthermore, this study was limited by the sample size, making stratified analyses of EBV positivity susceptible to type II errors.

## 5. Conclusions

A significant proportion of 18.7% of GRC in this population-based study from the West was EBV-positive, compared to 6.0% in the proximal non-GRCs, OR 3.6 (95% CI 1.4–9.3). Patients with GRC did more often present with GI bleeding and less often with upper abdominal pain. At endoscopy, an ulcer or a tumour was found in only 53% of the patients with GRC, significantly less than in the proximal non-GRCs, a subtlety that clinicians should be aware of. Long-term survival rates did not differ between patients with GRC and their controls, but the significantly higher median age of 79 years in the GRC group, combined with the aforementioned findings, warrants gastric remnant cancer to be designated a distinct disease entity with the Epstein–Barr virus as a causative contributor [5].

## Figures and Tables

**Figure 1 cancers-16-02000-f001:**
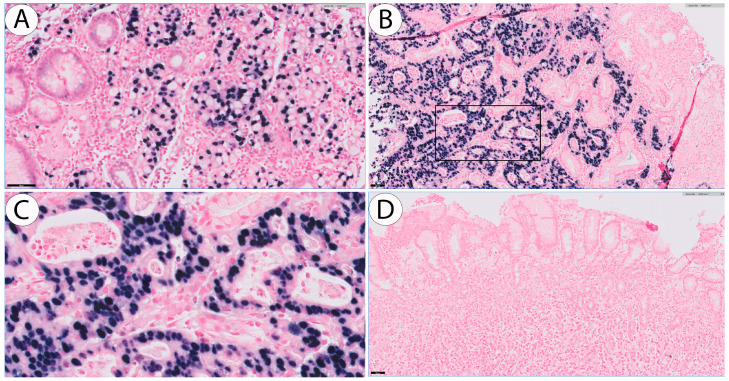
EBV ISH-positive cancer of Lauren diffuse type (**A**) and cancer of Lauren intestinal type (**B**). Positive tumour cells which are dark purple in colour are illustrated at a higher magnification in (**C**), i.e., the area marked by the rectangle in (**B**), where the majority of tumour cells are positive for EBV. The negative control sections did not have any staining, as seen in (**D**). Scale bar is 50 µm.

**Figure 2 cancers-16-02000-f002:**
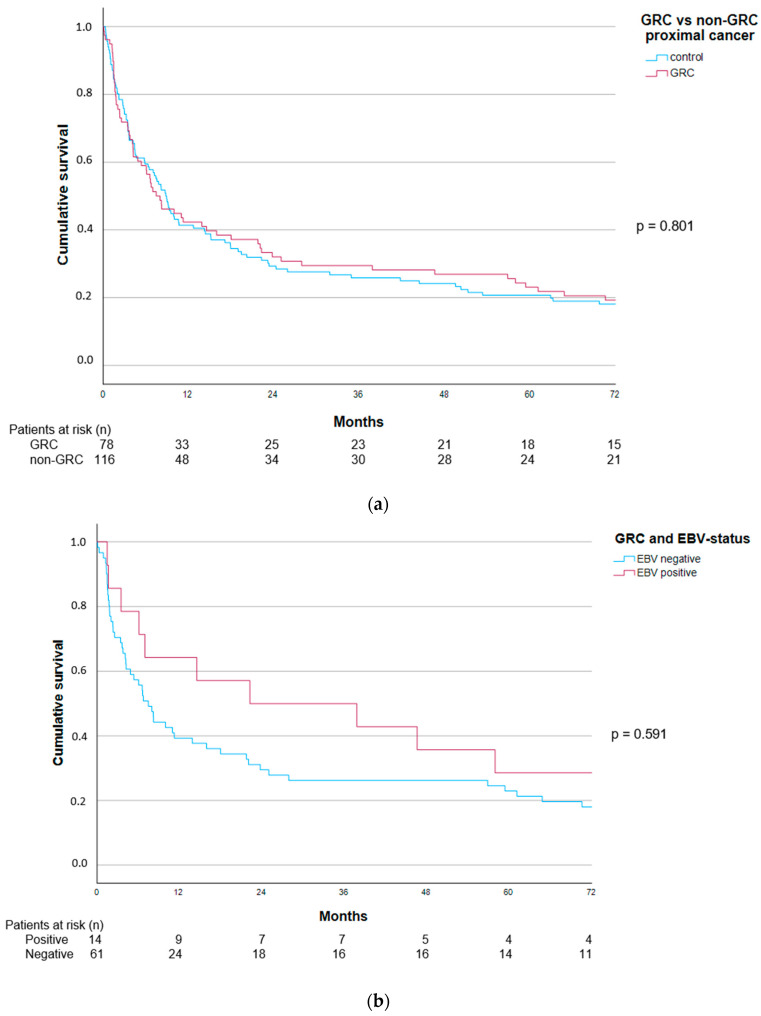
(**a**) Overall survival in patients with GRC (n = 78) and proximal non-GRC (n = 116), log-rank *p* = 0.801. (**b**) Overall survival in patients with GRC (n = 75) stratified by tumour EBV status. EBV-positive (n = 14), EBV-negative (n = 61), log-rank *p* = 0.591.

**Table 1 cancers-16-02000-t001:** Patient and tumour characteristics of GRC vs. proximal non-GRC (controls), n (%).

Variable	GRC, n = 78	Proximal Non-GRC, n = 116	*p*-Value
**Median age, years (range)**	79 (52–95)	72 (38–93)	0.005
**Male sex**	65 (83.3)	81 (69.8)	0.033
**(y)pTNM stage**			0.250
St 0 + I	18 (23.1)	12 (10.3)	
St II	5 (6.4)	18 (15.5)	
St III	7 (9.0)	13 (11.2)	
St IV + X	31 + 17 (61.5)	57 + 16 (62.9)	
**Chemotherapy**			
Perioperative	2 (2.6)	23 (19.8)	<0.001
Palliative	12/78 (15.4)	39/116 (33.6)	0.005
**Treatment**			
R0/R1 resection	32 (41.0)	48 (41.4)	0.961
R2 resection	4 (5.1)	-	
Local resection	-	3 (2.6)	
Non-resection interv. *	8 (10.3)	7 (6.0)	
No surgical intervention	34 (43.6)	58 (50.0)	
**Lauren classification**			0.914
Intestinal	44 (56.4)	61 (52.6)	
Diffuse	18 (23.1)	31 (26.7)	
Mixed	9 (11.5)	12 (10.3)	
Unspecified	7 (9.0)	12 (10.3)	

* GRC group: 1 gastro-jejunostomy, 7 explorative laparotomies. Non-GRC group: 2 gastro-jejunostomy, 2 explorative laparotomy, 3 endoluminal stent.

**Table 2 cancers-16-02000-t002:** Epstein–Barr virus in situ hybridization (ISH) analysis of tumour tissue samples from GRC (n = 75 *) and proximal non-GRC (controls) (n = 116), n (%).

Variable	EBV Positive	EBV Negative	*p*-Value
**Tumour location**			0.006
GRC	14 (18.7)	61 (81.3)	
Proximal non-GRC	7 (6.0)	109 (94.0)	
**Sex**			0.244
Male	18 (12.5)	126 (87.5)	
Female	3 (6.4)	44 (93.6)	
**Lauren classification**			0.873
Intestinal	13 (12.5)	91 (87.5)	
Diffuse	4 (8.2)	45 (91.8)	
Mixed	2 (9.5)	19 (90.5)	
Unspecified	2 (11.8)	15 (88.2)	
**Age category**			0.478
≤50 years	0 (0.0)	7 (100.0)	
51–70 years	8 (14.0)	49 (86.0)	
≥71 years	13 (10.2)	114 (89.8)	

* Only 75/78 GRC with sufficient tissue for EBV in situ hybridization (ISH).

**Table 3 cancers-16-02000-t003:** Symptoms at time of diagnosis and findings at upper endoscopy (UE) in GRC vs. proximal non-patients with GRC (controls), n (%). Each patient could have several entries.

Variable	GRC	Proximal Non-GRC
**Indication for UE**	n = 78	n = 116
Surveillance UE	4 (5.1)	5 (4.3)
Referred for symptoms	67 (85.9)	107 (92.2)
Missing data	7 (9.0)	4 (3.4)
**Symptoms** *****	n = 71	n = 112
None	4 (5.6)	5 (4.5)
Dyspepsia/dysphagia	23 (32.4)	48 (42.9)
GI-bleeding	22 (31.0)	18 (16.1)
Vomiting	12 (16.9)	20 (17.9)
Weight loss/general symptoms	34 (47.9)	55 (49.1)
Anaemia	25 (35.2)	25 (22.3)
Abdominal pain	10 (14.1)	40 (35.7)
**Findings at UE** ******	n = 69	n = 110
Normal ***	4 (5.8)	0 (0.0)
Gastritis	10 (14.5)	10 (9.1)
Ulcer	14 (19.7)	42 (38.2)
Tumour/polyp	29 (40.8)	73 (66.4)
Friable bleeding tissue	17 (24.6)	25 (22.7)
Other	13 (18.8)	8 (7.3)

* Data missing for 7 (9.0%) Patients with GRC and 4 (3.4%) controls. ** Data missing for 9 (11.5%) patients with GRC and 6 (5.2%) controls. *** Diagnosed by random biopsies from gastro-enteric anastomosis.

## Data Availability

Patient consent was waived due to the retrospective nature of the study and a long time span, causing the large majority of the patients to be inaccessible. The project data cannot be shared according to regulations given by the Regional Committee for Medical and Health Research Ethics, unless a specific application is forwarded. This study used data from the Cancer Registry of Norway. The interpretation and reporting of these data are the sole responsibility of the authors, and no endorsement by the Cancer Registry of Norway is intended nor should be inferred. Data from the Norwegian Patient Register were used in this publication. The interpretation and reporting of these data are the sole responsibility of the authors, and no endorsement by the Norwegian Patient Register is intended nor should be inferred.

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
