# Peer review of "Epstein–Barr Virus and Clinico-Endoscopic Characteristics of Gastric Remnant Cancers Compared to Proximal Non-Remnant Cancers: A Population-Based Study"

_cancers, 2024, doi:10.3390/cancers16112000_

Round 1

Reviewer 1 Report

Comments and Suggestions for Authors

Descriptive paper that brings to the general attention of surgeons, oncologists and gastroenterologists a problem that often, due to its limited diffusion in the population, passes into the background. Indeed, in numerous centers that specifically deal with gastric neoplasia, lesions of the upper stomach wall occur but due to the very low frequency of EBV gastric cancer, the latter is not looked for. Even tumors of the stump, after gastric resection many years earlier, are attributed to the action of the bile rather than to the possible presence of the virus in question. However, the issue must not be seen as just a transitory note because from a therapeutic point of view we could treat patients with immunotherapy doi: 10.1136/jitc-2021-004080. DOI: 10.1158/1078-0432.CCR-21-3408 and this could make the difference, Bibliography to be increased, good iconography, English to be revised

Comments on the Quality of English Language

English needs improvement

Author Response

First, we would like to thank the reviewers for a positive reception of our manuscript entitled “Epstein-Barr Virus and Clinico-endoscopic Characteristics of Gastric Remnant Cancers Compared to Proximal Non-remnant Cancers. A Population-based Study”, and several suggestions for improvements. Please find below a point-by-point response to the reviewers. Any changes made in the manuscript have been highlighted in red to facilitate tracking. Changes made to reference style, adding of references, pictures, and so on have not been marked in the manuscript per se, only in the running text when relevant.  

Reviewer 1

Descriptive paper that brings to the general attention of surgeons, oncologists and gastroenterologists a problem that often, due to its limited diffusion in the population, passes into the background. Indeed, in numerous centers that specifically deal with gastric neoplasia, lesions of the upper stomach wall occur but due to the very low frequency of EBV gastric cancer, the latter is not looked for. Even tumors of the stump, after gastric resection many years earlier, are attributed to the action of the bile rather than to the possible presence of the virus in question. However, the issue must not be seen as just a transitory note because from a therapeutic point of view we could treat patients with immunotherapy doi: 10.1136/jitc-2021-004080. DOI: 10.1158/1078-0432.CCR-21-3408 and this could make the difference, Bibliography to be increased, good iconography, English to be revised

Answer: Thank you for sharing with us the stance that gastric remnant cancer and EBV status is of importance. We agree that immunotherapy is a promising treatment modality, offering new angles of attack in treating the EBV positive subset of gastric cancers. A sentence has now been added in the discussion (lines 303-304), and the two references offered have  been included in the manuscript. To enhance bibliography, additional references have been added in other sections of the manuscript as well.

For English revision, the manuscript has now been thoroughly revised by a native English speaking colleague. 

Reviewer 2

1- Retrospective design may lead to bias, especially in recording symptoms and endoscopic findings. This should be clarified and the symptoms and endoscopic findings should be detailed for readers to understand clearly.

Answer: We fully agree with the warning on bias, in particular when registering variables such as symptoms and findings at UE. This was explicitly stated in the original manuscript under limitations. To further underscore this point a new sentence has been added in the running manuscript (lines 288-89). The study dates back to pre-routine imaging at endoscopy, and we have no possibility to illustrate specific findings at a macroscopic level by original illustrations. The description of the various types of findings did not allows us to present endoscopic findings at a more detailed level.

2a- The relationship between GRC and Proximal non-GRC in the table should be explained on the basis of the literature and the compatibility for the table should be discussed.                                                                                                Answer: when comparing GRC with a control group many previous studies have chosen proximal non-remnant cancers and there are several arguments for this choice. Cancers in the antrum, corpus and cardia have different risk factors and patients have different characteristics. We have for instance previously found that hypergastrinemia is a risk factors for cancer in the corpus/fundus, but not for cancer in the antrum. More important, it has previously been found that the proportion of EBV positive gastric cancers in the proximal stomach is higher than in distal stomach. The rationale for choosing proximal non-GRC as controls have been mentioned in the revised manuscript in Methods section 2.1 and was written in the original Discussion section 4.1., which has been left unchanged.

2b- The sample size limitation of the study may affect the statistical power for stratified analyzes and lead to type II errors. In this sense, the sample size should be evaluated taking into account its statistical properties.                                        Answer: We fully agree that sample size and type II errors is an issue for the GRC cancers in particular.  This reflects the rarity of GRC in a Western cohort – with this study spanning 16 years to achieve 78 pts. The issue was already commented on in the original manuscript at several places (300, 301  -  317, 318). Keeping this in mind, we have limited the number of subset analyses, and whenever doing so, warned of type II errors. As we see it, not much more can be added to it in this retrospective setting.

3- The rows and columns of the values in Table 1 appear mixed, making it difficult to read the table. This problem must be resolved.                                                                                                         Answer: We have now changed the denominator in the lines with the entries on palliative chemotherapy, to maintain the same denominator as for the other entries. The subtitle of Table 1 was changed accordingly and a new p-value calculated, without any change to the conclusions. As far as we can see, this was the possible source of number confusion. We cannot find any mix up between numbers and rows in table 1.

 4- Since the Findings at UE, n (%) values in Table 3 reflect the findings of the study, I recommend that these value ranges be explained in detail with their limit values.                                                                                                                  Answer: Admittedly, we are not certain of what is asked for by the reviewer at this point. Every patient might have more than one finding at UE (thus adding up to more than 100%), and this is merely registered and analyzed as a nominal variable. No number value or ranging of severity as in an ordinal variable was attempted, and hence, no range in numerical values or limit values can be offered. Considering the nominal variable, we find that relevant categories are adequately represented, covering the findings of any substantial findings at UE.

5- Figure 1 should be made more descriptive and enriched by using more explanations and additional figures. Additionally, Figure 2 needs to be detailed for a better understanding of the study.                                                             Answer: We thank the reviewer for pointing this out. Figure 1 has now been enriched by adding two more pictures, detailing some of the findings at a higher resolution as well as adding a negative control. The figure legend has also been expanded and we hope this has increased the readability for the manuscript.  In Figure 2 the number of patients at risk at each timepoint have been added below the horizontal axes, which we believe gives a better and more intuitive understanding of the figure.

6- In the results and discussion section, the contribution of the study to the literature should be emphasized with a sentence, and the comparison of the study with other studies should be included in the discussion section.                                                                                                                                            Answer: As stated in the manuscript, most of the reported data are from the East, with limited relevant literature from the West to compare with. This has been communicated throughout the manuscript (e.g. Introduction line 81 and Discussion lines 248,249). In addition, we have now added a sentence in the Discussion, clearly highlighting the novelty of our results from a Western perspective (line  242).

As for comparison of results, in the Discussion we have chosen to compare to results from meta-analyses, rather than individual studies, due to the rarity of individual studies, the small number of GRCs in each study, and the large variation in EBV status reported, e.g line 236.

Reviewer 3

This is a very important documentation on Norwegian gastric cancer in remnant stomach. May I ask the authors address following points?

  1. If available the list (table) of the reasons for distal gastrectomy would be helpful. Are there any stomach tissue left at gastrectomy?                        Answer: The reason for (the index) distal gastrectomy was stated in the original manuscript section 3.1 (line 144). That is: only two patients had an index operation for malignant disease, the remaining for intractable/recurrent ulcer disease. Hence, all patients had the proximal stomach remaining after index surgery.

  1. To my understanding, majority of population got EBV in teens in developed countries, thus the author think residence of EBV in stomach cancer implies just as passengers in stomach tissue in people who got operations? Answer: As pointed out by the reviewer, EBV infection occurs in most children and adolescents, infects B-cells and epithelial cells and thereafter the virus resides inside B-cells.  The exact mechanism by which EBV causes cancer from here is unknown, but there is monoclonal proliferation of EBV positive cells in a varying proportion of gastric cancers.  To what extent EBV may also reside in gastric epithelial cells after primary infection is uncertain, but the tumour- adjacent gastric mucosa contains none or very few EBV positive epithelial cells.  Some changes have been done in the manuscript to highlight these aspects.

  1. The authors indicated ISH in these positive cases, but histopathological features (as Burke first reported "lympho-epithelioma" ) were recognized? Answer: Lymphoepithelioma-like gastric carcinoma (LELC) is a type of Epstein-Barr virus (EBV)-associated gastric cancer, characterized by the presence of a lymphoid stroma with cells arranged primarily in microalveolar, thin trabecular and primitive tubular patterns, or isolated cells. Our pathologist (PM) has considered the question, however no such distinctive pathologic pattern was observed among the 78 GRCs as they were classified according to Lauren. This was added to the revised manuscript. It should be noted that cancers with LELC morphology are rare. 

  1.  Any MSI conducted in positive cases? Answer: No MSI assessment was done for this particular study. As it happens, our research group in a previous study (unpublished data) did find a MSI frequency of around 5% for a subset in Central Norway cohort of patients 75 years or below and with curable disease. This is at present the only assessment of MSI frequency in our regional database. An EBV positivity of 15% and an MSI frequency below 10% indicate that combined numbers and sub-analyses would unfortunately not be informative.

  1. Is it possible to assume helicobacter infection rate in Norwegian gastric ulcer subjects? How about in these positive and negative/ remnant vs non-remnant cases? Answer: Although an interesting question, we do not have Helicobacter pylori status in our patient cohort. It is reasonable to assume, however, that a large majority of the GRC cancers at some time prior to the index operation was Helicobacter positive, since 76/78 of the GRC patients had their index operation for recurrent and intractable peptic ulcer disease.

  1. It is famous Norwegian national wide database to follow various diseases. Is possible basic data on gastrectomy cohort (total numbers a decade, age at operation,  and others). Are there any EBV antibody profile in the population during this periods? Answer: As correctly alluded to, The Norwegian Cancer Registry has been at work since the early 1950s. It is, however, a database restricted to new cancer entities, and carries only limited information on epidemiological variables and no blood-samples have been stored. There are for instance no registration of GRC in particular at a national level. To the best of our knowledge, there is no available data on EBV positivity for the general population in Norway during the relevant time-period, but it is assumed to be at a level above 90%.
  2. On survival curves, how much operable cases in there? OR any stages recorded in the discovery of remnant GC? Answer: Number of resectable cases and so on was stated in Table 1, alongside with stage distribution for both the GRC group and the control group. For survival curves, if numbers at risk was what was asked for, these are now provided in the revised Figure 2.

Reviewer 2 Report

Comments and Suggestions for Authors

This study examines the distinct nature of EBV-associated gastric cancer, focusing specifically on GRC that occurs years after distal gastrectomy. While most data on this topic originate from the East, this study examines a Western cohort from Central Norway (2001-2016). Using a population-based approach, it reveals that even in the West, EBV positivity is significantly higher in GRC than in proximally located cancers without GRC. GRC presents with more subtle symptoms and endoscopic findings, leading to diagnostic difficulties. In this study, the authors emphasize that GRC should be considered a separate disease entity to which EBV contributes. In this sense, the study addressed an important specific issue. However, I bring to the authors' attention the following suggestions for improvement.

1- Retrospective design may lead to bias, especially in recording symptoms and endoscopic findings. This should be clarified and the symptoms and endoscopic findings should be detailed for readers to understand clearly.

2- The relationship between GRC and Proximal non-GRC in the table should be explained on the basis of the literature and the compatibility for the table should be discussed.

2- The sample size limitation of the study may affect the statistical power for stratified analyzes and lead to type II errors. In this sense, the sample size should be evaluated taking into account its statistical properties.

3- The rows and columns of the values in Table 1 appear mixed, making it difficult to read the table. This problem must be resolved.

4- Since the Findings at UE, n (%) values in Table 3 reflect the findings of the study, I recommend that these value ranges be explained in detail with their limit values.

5- Figure 1 should be made more descriptive and enriched by using more explanations and additional figures. Additionally, Figure 2 needs to be detailed for a better understanding of the study.

6- In the results and discussion section, the contribution of the study to the literature should be emphasized with a sentence, and the comparison of the study with other studies should be included in the discussion section.

Author Response

(The authors gave the same response as above.)

Reviewer 3 Report

Comments and Suggestions for Authors

This is a very important documentation on Norwegian gastric cancer in remnant stomach. May I ask the authors address following points?

1. If available the list (table) of the reasons for distal gastrectomy would be helpful. Are there any stomach tissue left at gastrectomy?

2. To my understanding, majority of population got EBV in teens in developed countries, thus the author think residence of EBV in stomach cancer implies just as passengers in stomach tissue in people who got operations?

3. The authors indicated ISH in these positive cases, but histopathological features (as Burke first reported "lympho-epithelioma" ) were recognized?

4.  Any MSI conducted in positive cases?

5. Is it possible to assume helicobacter infection rate in Norwegian gastric ulcer subjects? How about in these positive and negative/ remnant vs non-remnant cases?

6. It is famous Norwegian national wide database to follow various diseases. Is possible basic data on gastrectomy cohort (total numbers a decade, age at operation,  and others). Are there any EBV antibody profile in the population during this periods?

7. On survival curves, how much operable cases in there? OR any stages recorded in the discovery of remnant GC?

Author Response

(The authors gave the same response as above.)
